# Coal Mine Safety Accidents, Environmental Regulation and Economic Development—An Empirical Study of PVAR Based on Ten Major Coal Provinces in China

**Cheng Lu** [1], **Shuang Li** [1,2,*], **Kun Xu** [1,2] and **Jiao Liu** [1,2]

[1] Safety Science and Emergency Management Research Institute, China University of Mining and Technology, Xuzhou 266525, China
[2] School of Economics and Management or School of Management, China University of Mining and Technology, Xuzhou 266525, China
* Correspondence: lishuangchina@cumt.edu.cn

**Abstract:** Based on the PVAR model and taking the data of 10 major coal provinces in China from 2011 to 2020 as an example, the dynamic relationship between coal mine accidents, environmental regulation and economic development is analyzed at the provincial level. Research findings include: (1) From the static relationship between coal mine accidents, environmental regulation and economic development in China's ten major coal provinces, coal mine accidents promote environmental regulation; environmental regulation inhibits coal mine accidents; economic development strongly promotes environmental regulation; environmental regulation has a weak inhibitory effect on economic development; coal mine accidents slightly inhibit economic development; economic development strongly inhibits coal mine accidents. (2) From the dynamic relationship between coal mine accidents, environmental regulation and economic development in China's ten major coal provinces, there is a strong dynamic response relationship between environmental regulation and coal mine accidents. The impact of environmental regulation on coal mine accidents shows a decreasing volatility trend, and the impact of coal mine accidents on environmental regulation shows a rising volatility trend. There is a short-term positive interaction between economic development and environmental regulation, but the interaction response relationship between them decreases with time. Economic development has a long-term inhibitory effect on coal mine accidents, while the negative impact of coal mine accidents on economic development has gradually decreased to 0.

**Keywords:** coal mine accidents; environmental regulation; economic development; PVAR

## 1. Introduction

Economic growth is the result of human efforts to transform nature, conquer nature and create wealth [1]. The utilization and transformation of coal as a natural resource produces huge economic benefits and promotes economic and social development. However, accidents at mine sites provide an area of concern, particularly considering that the coal mining operations and economic growth are linked to the social production system.

The resource distribution conditions of relatively rich coal deposits but poor oil and gas reserves in China dictate that coal will still be the main source of energy in China for a long time in the future, but China's 2030 target of carbon peaking requires adjusting the energy structure and building a clean energy system. Under the environmental regulation of carbon peaking, the contradiction between safe coal production and economic growth is becoming an increasingly prominent issue which could lead to more severe mining accidents and cause economic and human losses.

Studying the dynamic relationship between coal mine accidents, environmental regulation and economic development can help us understand the push–pull factors of coal

production under the dual carbon target. Coal mining must be safe to help ensure sustainable economic growth. Environmental regulation is a long-term requirement for the sustainable development of the coal industry. Therefore, exploring the dynamic relationship between coal mine accidents, environmental regulation and economic development and understanding the endogenous relationship and mechanism of the three are conducive to providing suggestions and countermeasures for safe coal mine production and economic growth under the environmental regulation of carbon peaking.

## 2. Literature Review

### 2.1. Relationship between Economic Development and Coal Mine Accidents

The theory of safety economics holds that the current situation of safe production reflects the social and economic development level of the country and the management ability of governments [2]. Often, in the drive to industrialize, developing nations with per capita GDPs below USD 5000 accept less safety in all industries, including coal mining. As a result, coal mine accidents fluctuate and increase. When per capita GDP reaches about USD 10,000, the coal mine accidents can reach a steady decline as governments and companies work to increase safety regulations. When per capita GDP is above USD 20,000, coal mine accidents are usually well controlled [3]. The relationship between mining accidents and economic development in China also reflects a trend: when the economic growth rate is significantly accelerated and the social system has undergone major changes, the number of coal mine accidents significantly increase; when the economic growth rate decreases and the social system is stable, the number of coal mine accidents decrease significantly and tend to be stable [3].

In previous studies, scholars have conducted a lot of analysis on the relationship between mining accidents and economic development in the world and in China. Huang et al. compared and analyzed the relationship between the death rate per 100,000 people and the per capita GDP in 27 sample countries in 1990 and 2000 and found an interesting rule: the overall safety of production improves with economic and social development. However, China's economy is rapidly developing, yet the number of accident deaths in China is on the rise [4]. Liu et al. found that when economic growth accelerated, the death toll from coal mine accidents would increase significantly, and when economic development was stable, the death toll of coal mine accidents would decline and maintain a stable trend [5]. Chen studied the relationship between regional economies and coal mine accidents in western, central and eastern China from 2001 to 2010 through a Lorentz curve analysis. The study found that regional economic imbalance led to a huge gap in the death rate from coal mine accidents, with the death rate in western and central China 14.8 times and 8.5 times higher than that in eastern China, respectively [6].

The changing characteristics of foreign and domestic coal mine safety accident can reflect the relationship between social economy and safety production level. The level of safety production shows different performance in different stages of social and economic development.

### 2.2. Relationship between Environmental Regulation and Coal Mine Accidents

From the perspective of regulatory economics, the development of shared resources by a large number of private operators will inevitably lead to the "tragedy of shared land", and the government must regulate the behavior of these private operators [7]. Most people who hold the view of "regulation" attribute the cause of "mine disaster" to "weak regulation", including high regulation cost (especially information cost), corruption of local officials, etc. [7]. Under the dual carbon target, the environmental regulation on coal enterprises is unprecedented, and a large number of outdated production methods need to be replaced. Closing mines and stopping production has become the strongest punitive environmental regulation policy for noncompliant mining operations. Bai et al. used the provincial parallel data of township coal mines from 1995 to 2005 to test the impact of the "mine closure policy" on coal mine accidents using difference-in-difference. The results showed that the "mine

closure policy" significantly increased the death rate of coal mine accidents [8]. The above analysis shows that the overly tough punitive environmental regulations are not conducive to improving coal mine safety, and there are two situations leading to the rise of coal mine accidents. The first is that if coal mining enterprises cannot bear the high cost of regulation and choose to evade regulation, there will be a sharp increase in accidents in the short-term. The second is that the "mine closure policy" reduces the coal supply, pushes up the price (or at least curbs the decline in coal prices) and leads to increased production of the coal mines that have not been closed. As a result, the accident rate may rise. Under a relatively moderate environmental regulation policy, such as the incentive environmental regulation policy, government and coal mine safety have a different development relationship. Xiang used a VAR model to analyze China's incentive environmental regulation policy (pollution control investment) and coal mine safety from 1985 to 2016, and the conclusion was drawn that the policy can help reduce coal mine accidents, improve overall mine safety and also motivate government environmental regulation [9].

The above analysis shows the very different impacts that punitive and incentive environmental regulation have on mining safety in the short-term.

### *2.3. Review*

To sum up, domestic and foreign scholars have conducted a lot of research on coal mine accidents, environmental regulation and economic development, but there are still some problems worth discussing. First, the research on coal mine accidents and economic development was concentrated in China from 1990 to 2010, which mainly verified the theory that the per capita GDP was less than USD 5000, and the rapid economic development made coal mine accidents fluctuate and increase. There is a lack of theoretical research on the steady decline of coal mine accidents due to stable economic development with a per capita GDP above USD 5000. Second, previous studies focused on the one-way impact of the two dimensions of environmental regulation—coal mine accidents and economic development—lacking the interaction between coal mine accidents, environmental regulation and economic development. Therefore, this paper uses the provincial panel data of China's ten major coal provinces from 2011 to 2020 to establish a PVAR model of coal mine accidents, environmental regulation and economic development to explore the dynamic response relationship of the three.

### 3. Methodology

#### *3.1. Econometric Model*

This paper adopts the panel vector autoregression (PVAR) model proposed by Love and Zicchino. The PVAR model has several econometric advantages to make it the best method to test macroeconomic dynamics [10]. First, PVAR is helpful in analyzing the impact propagation between variables in unit time. Second, PVAR is based on the analysis of real data series, rather than adhering to the concept of macroeconomics. Third, the model does not lead to differences between dependent and independent variables but regards all factors as mutually endogenous. In addition, it also provides the interactive response process of dependent variable and independent variable. The PVAR model is shown in Equation (1):

$$y_{it} = \beta_0 + \sum_{j=1}^{k} y_{i,t-j} + \gamma_i + \delta_t + \varepsilon_{it} \tag{1}$$

In Equation (1), $y_{it}$ is a column vector containing 3 variables, $i$ and $t$ represent provinces and time, respectively, $\beta_0$ represent intercept items, $\beta_j$ represent coefficient matrix of lag order $j$, $\gamma_i$ represents the regional fixed effect, $\delta_t$ represents the time fixed effect, and $\varepsilon_{it}$ represents the stochastic error. When discussing the relationship between mining accidents, environmental regulation and economic development, $y_{it}$ is a three-dimensional endogenous variable vector.

*3.2. Variable Description*

(1)　Coal mine accidents

In this paper, the death rate per million tons of coal (CD) is selected to represent the level of coal mine accidents. CD reflects the coal mine safety production guarantee capacity, and it is an important indicator to measure the severity of accidents.

(2)　Economic development

GDP is often used as a measure of the level of economic development, but in order to avoid the vertical incompatibility of economic data caused by regional population differences, this paper chooses per capita GDP (PCGDP) to represent the level of economic development.

(3)　Environmental regulation

The indicators selected for measuring the intensity of environmental regulation (ER) in the previous literature are different. They are mainly divided into the following categories: (1) The previous literature used pollutant emissions as environmental pollution indicators to measure the intensity of environmental regulation, such as SO2, NO2 emissions, industrial wastewater emissions, smoke and dust emissions, etc. [11]. (2) In previous studies, the number of environmental pollution control policies and the frequency of environmental protection words in government work reports were used to measure the intensity of environmental regulation [12]. (3) In previous studies, macroeconomic indicators such as environmental fiscal expenditure and environmental pollution control investment were selected to represent environmental pollution input indicators that measure the intensity of environmental regulation [13]. (4) In previous studies, qualitative indicators such as "mine closure" were selected to measure the intensity of punitive environmental regulation indicators [8].

Because the PVAR model does not distinguish between dependent variables and independent variables, and all factors are regarded as mutually endogenous, policy differences need not be considered. Township coal mines have mostly vanished, and "mine closure policy" is difficult to enforce in existing coal mines, so punitive environmental regulations are no longer considered. It is difficult to obtain the environmental pollution policy indicator data. It is therefore necessary to use alternative indicators in the robustness test of the model, which easily cause deviation. Therefore, the environmental pollution policy indicators will not be considered. Instead, this paper uses environmental pollution input indicators to measure the intensity of environmental regulation, considering the economic differences in different regions. The final environmental regulation indicator is the proportion of environmental protection expenditure and the proportion of industrial pollution control investment.

Because the two indicators are proportional indicators, the differences in indicator characteristics lead to their incommensurability. First, the indicator should be standardized, and then the intensity of environmental regulation can be obtained through linear weighting. The steps are as follows:

Use Equation (2) to standardize indicators:

$$R_{ij}{}^+ = \frac{R_{ij} - R_{\min}}{R_{\max} - R_{\min}} \tag{2}$$

In Equation (2), $R_{ij}$ represents the original value of $j$ indicator of city $i$, $R_{\max}$ and $R_{\min}$ represent the maximum and minimum values of $j$ indicator in 10 major coal provinces, and $R_{ij}^+$ represents the standardized value of $j$ indicator of city $i$.

The two environmental regulation indicators have the same importance in environmental regulation, so the two indicators are given the same weight. Therefore, the intensity of environmental regulation should be as shown in Equation (3):

$$E_{Ri} = \frac{(R_{i1} + R_{i2})}{2} \tag{3}$$

### 3.3. Data Sources

This paper takes the panel data of death rate per million tons of coal, environmental regulation intensity and per capita of GDP Shaanxi, Shanxi, Inner Mongolia, Xinjiang, Guizhou, Anhui, Henan, Shandong, Gansu and Yunnan from 2011 to 2020 as samples. The sample data are all from *China Statistical Yearbook, China Safety Production Statistical Yearbook, Shaanxi Statistical Yearbook, Shanxi Statistical Yearbook, Inner Mongolia Statistical Yearbook, Xinjiang Statistical Yearbook, Guizhou Statistical Yearbook, Anhui Statistical Yearbook, Henan Statistical Yearbook, Shandong Statistical Yearbook, Gansu Statistical Yearbook* and *Yunnan Statistical Yearbook* from 2012–2021 (Table 1).

**Table 1.** Descriptive statistical results.

| Variables | Number of Samples | Mean | Std. Dev. | Max | Min |
|---|---|---|---|---|---|
| CD | 100 | 0.234 | 0.322 | 0.009 | 1.838 |
| ER | 100 | 0.332 | 0.142 | 0.058 | 0.819 |
| PCGDP | 100 | 44,231.53 | 15,305.26 | 16,413 | 76,267 |

## 4. Empirical Analysis

### 4.1. Unit Root Test

Before building the model, it is necessary to conduct a unit root test on the time series data of each variable to avoid spurious regression in subsequent regression analysis. First, the variables are logarithmically processed, and then the Levin, Lin and Chut * (LLC) test is performed on the variables. The results of the LLC test for the variables LnCD showed $p > 0.05$, which indicated that the data of the variables were non-stationary. In order to ensure that the variables were of the same order, the variables LnCD, LNER and LNPCGDP were processed by the first difference. The LLC test after the first difference showed $p < 0.05$ for all variables, which indicated that all variable data were stable and could be modeled (Table 2).

**Table 2.** Unit root test results (LLC test).

| | Horizontal Data | | | First Difference Data | | |
|---|---|---|---|---|---|---|
| Variable | LnCD | LnER | LnPCGDP | DLnCD | DLnER | DLnPCGDP |
| *p* | 0.0014 | 0.1235 | 0.022 | 0 | 0 | 0.0001 |
| Test results | Stationary | Non-stationary data | Stationary | Stationary | Stationary | Stationary |

### 4.2. Determining the Lag Order

Before building the PVAR model, the optimal lag order of the model should be determined. According to LR, FPE, AIC, SC and HQ criteria, the optimal lag order is determined as order 1 (Table 3).

**Table 3.** Selection of the lag order of the model.

| Lag | LR | FPE | AIC | SC | HQ |
|---|---|---|---|---|---|
| 0 | NA | 0.000211 | 0.051194 | 0.155911 | 0.092155 |
| 1 | 44.04633 * | 0.000130 * | −0.435347 * | −0.016479 * | 0.271505 * |
| 2 | 8.85349 | 0.000149 | −0.302394 | 0.430626 | −0.01567 |
| 3 | 13.09594 | 0.000156 | −0.264313 | 0.782859 | 0.145293 |

Note: * represents the optimal lag order under this criterion.

*4.3. Robustness Test*

The purpose of the robustness test is to ensure that the model is effective. The points in the unit circle in Figure 1 represent inverse roots of an AR characteristic polynomial. If these points fall in the unit circle, the model is stable. It can be seen from Figure 1 that the PVAR model is robust.

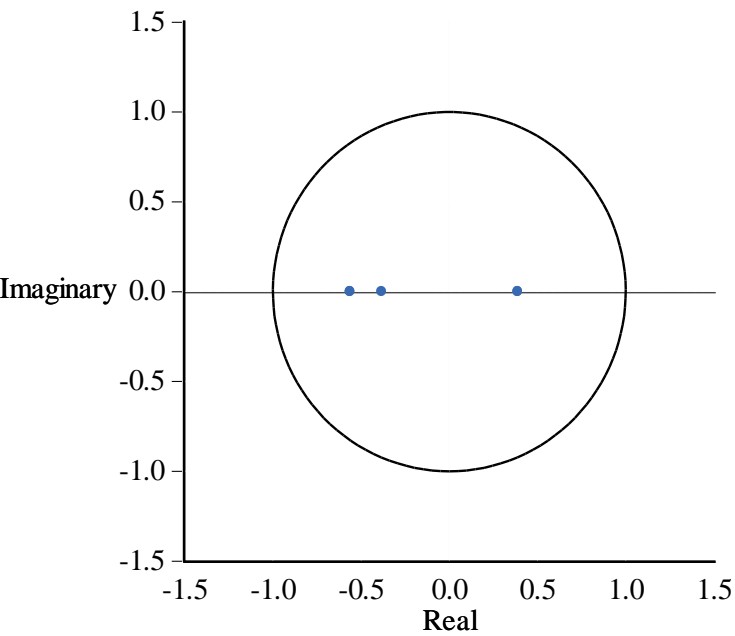

**Figure 1.** PVAR model robustness test result.

*4.4. GMM Estimation of the Model*

After the unit root test, determination of the optimal lag order and the robustness test, GMM estimation is further carried out for coal mine accidents, environmental regulation and economic development. The GMM estimation results of the model are shown in Table 4.

**Table 4.** GMM estimation results of the model.

| | DLNCD (a) | DLNEC (b) | DLNPCGDP (c) |
|---|---|---|---|
| DLNCD (−1) | −0.587197 [−5.68087] | 0.042065 [0.98721] | −0.014575 [−1.97144] |
| DLNEC (−1) | −0.175589 [−0.55404] | −0.338079 [−2.58778] | −0.001473 [−0.06497] |
| DLNPCGDP (−1) | −1.034829 [−0.70275] | 0.790727 [1.30263] | 0.372649 [3.53821] |

Note: The values in parentheses represent the corresponding *t*-statistics.

Model (a) with coal mine accidents as the explained variable showa that the influence coefficient of coal mine accidents on itself is −0.587, which indicates that coal mine accidents

have a strong self-restraining effect. The impact coefficient of environmental regulation on coal mine accidents is −0.176, which indicates that environmental regulation has a negative effect on coal mine accidents; that is, environmental regulation can inhibit coal mine accidents.

Model (b) with environmental regulation as the explanatory variable shows that the influence coefficient of environmental regulation on itself is −0.338, which indicates that environmental regulation has a strong self-restraint effect. The impact coefficient of coal mine accidents on environmental regulation is 0.04, which indicates that mining accidents can promote environmental regulation. The influence coefficient of economic development on environmental regulation is 0.791, which indicates that economic development has a strong promoting effect on environmental regulation.

Model (c) with economic development as the explained variable shows that the influence coefficient of economic development on itself is 0.373, which indicates that the self-promoting effect of economic development is obvious and has a "certain inertia". The influence coefficient of coal mine accidents on economic development is −0.015, which indicates that accidents will restrain economic development. The influence coefficient of environmental regulation on economic development is −0.0015, which indicates that environmental regulation has a weak restraining effect on economic development.

### 4.5. Impulse Response

The above GMM estimation is a static analysis of the model. In order to further conduct dynamic analysis on the interaction between coal mine accidents, environmental regulation and economic development, the impulse response is used to estimate the impact of one standard deviation of a random disturbance term on endogenous variables and analyze the influence of one variable on other variables when it changes in the base period. This section uses a Monte Carlo simulation to obtain the impulse response figure of ten lagging periods (Figures 2–5). In the figure, the abscissa represents the number of lag periods, the ordinate represents the degree of impulse response, and the middle solid line is the impulse response curve. The curves on both sides of the impulse response curve constitute a 95% confidence interval. Specific analysis leads to the following conclusions.

First, the response of DLNCD to DLNCD in Figure 2 shows the self-impact of coal mine accidents. When the coal mine accidents are impacted by themselves, they present a "W" trend which inhibits influence from other variables. The self-promoting effect is very strong at the first period, and tends to 0 in the fifth stage. The response of DLNER to DLNER in Figure 2 shows the self-impact of environmental regulation. When environmental regulation is impacted by itself, it has a strong positive promotion effect in the first period, then has a inhibition effect in the second period and is no longer affected by itself after the fourth period. The response of DLNPCGDP to DLNPCGDP in Figure 2 shows the impact of economic development by itself. When economic development is impacted by itself, it forms a long-term positive effect, which reaches the highest level in the first period, then begins to weaken, and approaches 0 in the fifth period. The economic development has maintained its self-promotion effect for five periods, which indicates that the economic development has a strong "inertia".

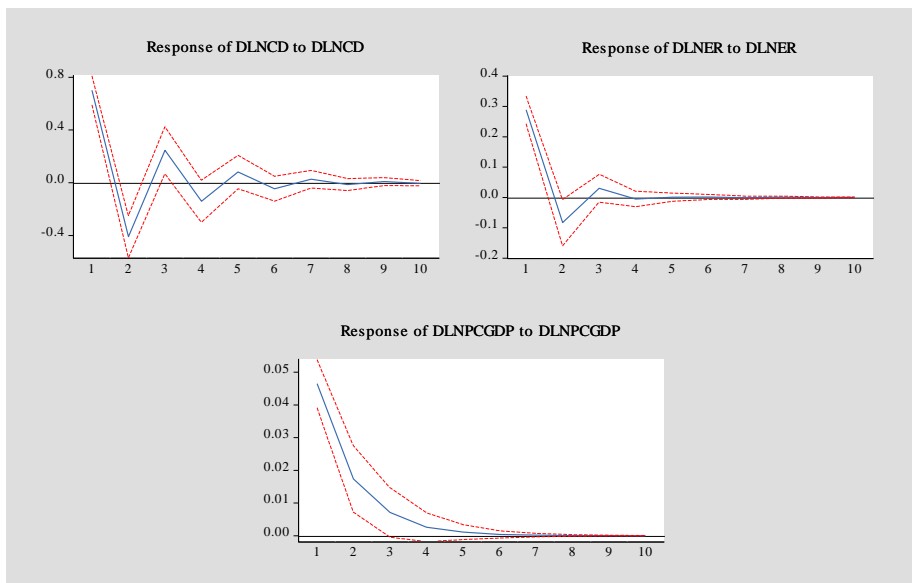

**Figure 2.** LNCD, LNER, LNPCGDP self-pulse response diagram.

Second, the response of DLNCD to DLNER of Figure 3 shows the impact of environmental regulation on coal mine accidents. Environmental regulation has no impact on accidents in the first period and then forms the trend of promoting and inhibiting the alternate impact in the second to the seventh period. The response of DLNER to DLNCD in Figure 3 shows the impact of coal mine accidents on environmental regulation. The impact of environmental regulation on accidents has a certain lag. Environmental regulation has a restraining effect on coal mine accidents from the second period, and then from the second period to the seventh period, it continues to show a trend of inhibiting and promoting the alternate impact.

The interaction between coal mine accidents and environmental regulation has lasted for seven periods, which shows that coal mine accidents and environmental regulation have a strong linkage relationship. The mutual dynamic response relationship between coal mine accidents and environmental regulation has just formed the opposite trend. When environmental regulation inhibits (promote) accidents, the accidents will promote (inhibit) environmental regulation. However, from the perspective of ten cycles of development, the impact of environmental regulation on mining accidents shows a decreasing volatility trend, and the impact of accidents on environmental regulation shows a rising volatility trend. Therefore, the interaction effect between coal mine accidents and environmental regulation is good as a whole but needs to be optimized.

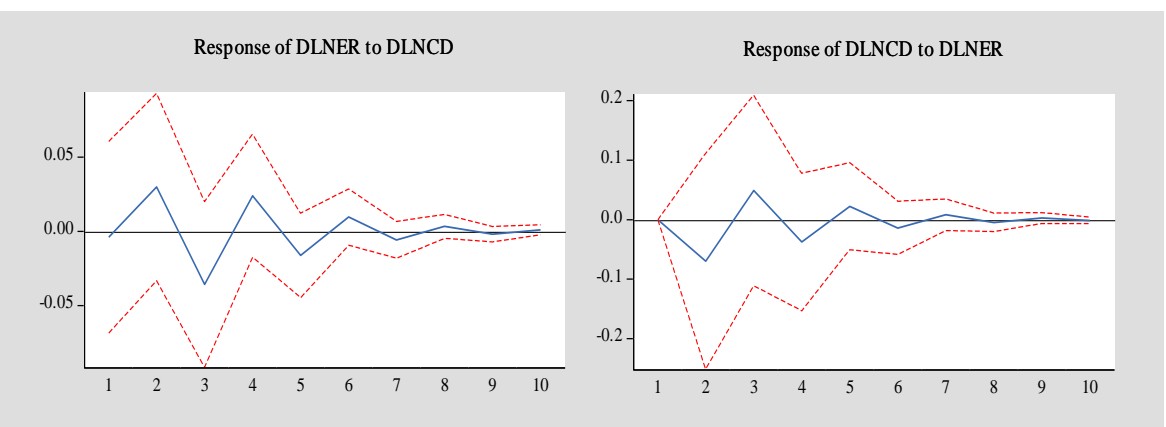

**Figure 3.** Mutual pulse response of LNCD and LNER.

Third, the response of DLNER to DLNPCGDP in Figure 4 shows the impact of economic development on environmental regulation. Economic development had no impact on environmental regulation in the first period but had a promoting effect in the second period, then fell back to no impact in the third period, had a weak promoting effect in the fourth period, and basically had no impact from the fifth to the tenth period. The response of DLNPCGDP to DLNER in Figure 4 shows the impact of economic development on environmental regulation. Environmental regulation has a long-term promoting effect on economic development, which reaches the highest level in the first period, continues to decline and disappears in the fourth to the tenth period.

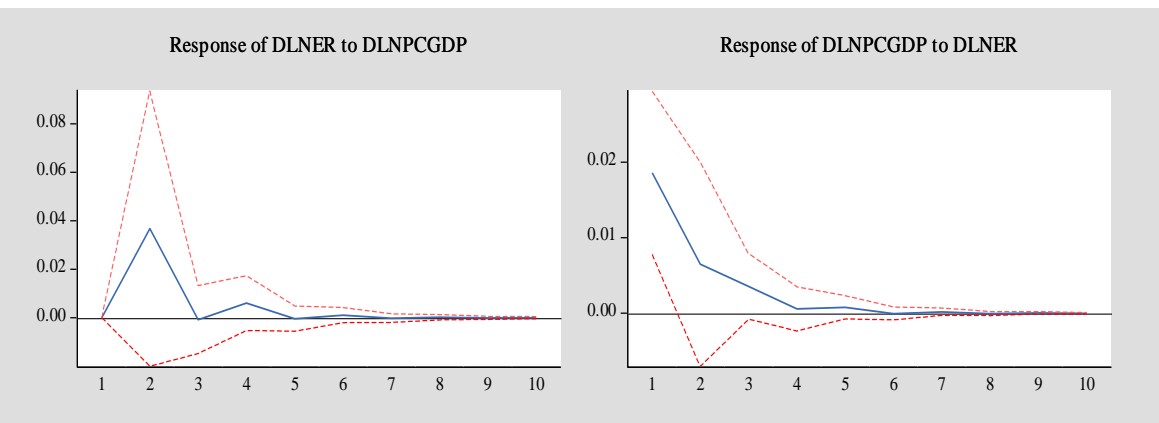

**Figure 4.** Mutual pulse response of LNER and LNPCGDP.

Economic development and environmental regulation have formed a good mutually promoting effect in the first four periods, which shows that a short-term positive interaction effect between economic development and environmental regulation has been established in the ten major coal provinces.

Fourth, the response of DLNCD to DLNPCGDP in Figure 5 shows the impact of economic development on coal mine accidents. When the economic development is impacted by a standard deviation of coal mine accidents, there is no impact effect in the first period, but negative impact in the second period, positive impact in the third period and negative impact in the fourth period. The response of DLNPCGDP to DLNCD of Figure 5 shows the impact of coal mine accidents on economic development. This trend of alternating positive and negative impact gradually weakens, and no impact will occur from the seventh period to the tenth period. Economic development has a long-term inhibitory effect on coal mine accidents, which reaches the highest level in the second period and has no impact in the fifth to tenth period.

Between the coal mine accidents and economic development, the interaction between accidents and economic development presents a situation of mutual inhibition, but the inhibition of accidents on economic development gradually decreases to 0, so the interaction between the two is more reasonable.

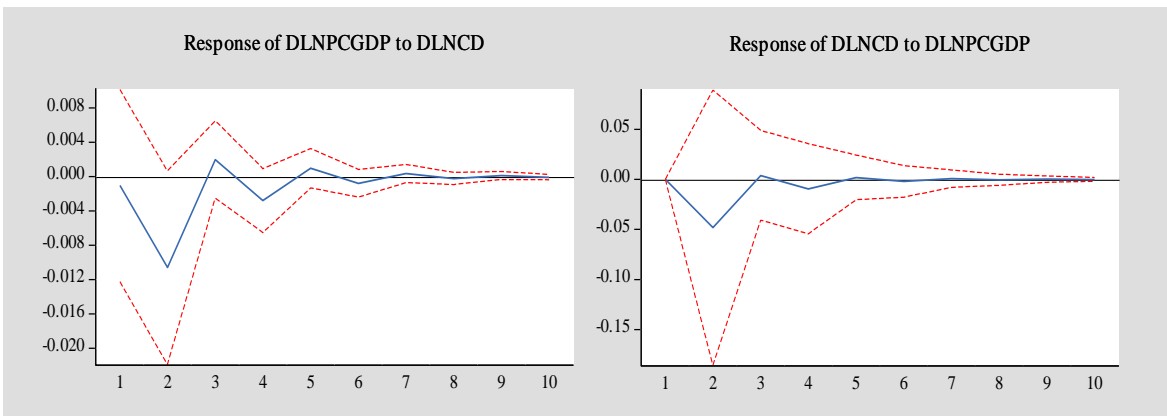

**Figure 5.** Mutual pulse response of LNCD and LNPCGDP.

*4.6. Variance Decomposition*

In order to determine the source of variance change of variables in the PVAR model, we analyze the impact of endogenous variables on the change of these variables and evaluate the importance of this impact. The variance decomposition analysis is used to explain the mechanism of coal mine accidents, environmental regulation and economic development in the PVAR model. The variance decomposition results are summarized in Table 5.

**Table 5.** Variance decomposition table.

| | DLNCD | | | DLNEC | | | DLNPCGDP | | |
| Period | DLNCD | DLNEC | DLNPCGDP | DLNCD | DLNEC | DLNPCGDP | DLNCD | DLNEC | DLNPCGDP |
|---|---|---|---|---|---|---|---|---|---|
| 1 | 100.00 | 0.00 | 0.00 | 0.02 | 99.98 | 0.00 | 0.04 | 13.78 | 86.18 |
| 2 | 98.92 | 0.73 | 0.35 | 0.99 | 97.55 | 1.46 | 3.82 | 13.09 | 83.09 |
| 3 | 98.68 | 1.00 | 0.32 | 2.32 | 96.26 | 1.43 | 3.86 | 13.22 | 82.92 |
| 4 | 98.52 | 1.16 | 0.32 | 2.90 | 95.64 | 1.46 | 4.10 | 13.17 | 82.73 |
| 5 | 98.46 | 1.22 | 0.32 | 3.17 | 95.37 | 1.45 | 4.13 | 13.17 | 82.70 |
| 6 | 98.44 | 1.24 | 0.32 | 3.27 | 95.28 | 1.45 | 4.15 | 13.17 | 82.68 |
| 7 | 98.43 | 1.25 | 0.32 | 3.30 | 95.25 | 1.45 | 4.15 | 13.17 | 82.68 |
| 8 | 98.43 | 1.25 | 0.32 | 3.31 | 95.24 | 1.45 | 4.15 | 13.17 | 82.68 |
| 9 | 98.43 | 1.25 | 0.32 | 3.32 | 95.23 | 1.45 | 4.15 | 13.17 | 82.68 |
| 10 | 98.43 | 1.25 | 0.32 | 3.32 | 95.23 | 1.45 | 4.15 | 13.17 | 82.68 |

The variance contribution of coal mine accidents mainly comes from itself, which remains at 98% in the tenth period, followed by environmental regulation and economic development. In the variance decomposition of environmental regulation, the self-variance contribution is maintained at more than 95%, followed by accidents and economic development. In the variance decomposition of economic development, its own variance contribution is 82–86%, the variance contribution of environmental regulation to economic development is maintained at 13%, and the variance contribution of accidents to economic development is maintained at 4%.

## 5. Discussion

From the perspective of China's actual development, the above empirical results are reasonable.

First, coal mine accidents promote environmental regulation, and environmental regulation inhibits coal mine accidents.

Previous scholars have shown that pollution, such as mine dust and noise, is an important factor causing coal mine accidents, and environmental regulation can reduce pollution in coal mines [14]. The output effect of environmental regulation supports the sustainable

development of the coal industry, rationalizes the production and consumption of coal, removes backward production capacity [15] and fundamentally improves the level of coal mine safety. Therefore, environmental regulation will inhibit coal mine accidents. However, at the initial stage of environmental regulation, the output effect of the regulations is limited, and the environmental regulation policy has not yet formed a long-term mechanism. Therefore, it is unable to inhibit accidents [16]. When environmental regulations produce certain effects, they will restrain the accidents. Controlling the accidents will increase the impact of environmental regulations further and increase the output of environmental regulations. So, the coal mine accidents promote environmental regulation.

Coal mine accidents and environmental regulation present the opposite trend of promoting and inhibiting each other. The main reason for this trend is that there is no stable environmental regulation output in the ten major coal provinces, so the control effect of coal mine accidents show a fluctuating trend. Of course, with stable economic development, environmental regulation will restrain accidents. Under this influence, coal mine accidents will show a trend of decreasing fluctuations. The government needs to formulate relatively mild environmental regulation policies, increase environmental investment and maintain stable environmental regulation to control coal mine accidents. Only in this way can environmental regulation maintain a stable inhibition on coal mine accidents and minimize the inhibition of coal mine accidents on environmental regulation.

Second, economic development strongly promotes environmental regulation; environmental regulation inhibits economic development.

There is a short-term positive interaction between economic development and environmental regulation. This interaction shows that reasonable environmental regulation can effectively stimulate or force coal enterprises to carry out technological innovation, achieve the task of energy conservation and emission reduction and improve the competitiveness and production efficiency of coal enterprises. The benefits of technological innovation compensate for the costs of pollution control investment. On the macro level, it can promote local economic development and improve the efficiency of economic development [17], but the effect of technological innovation brought by environmental regulation is limited. The energy and chemical industry of the ten coal provinces is significant, and the cost of environmental governance is high. Environmental regulation brought by economic development will play an important role in controlling pollution emissions in the early stage, but its effect on promoting structural upgrades and industrial technology progress is poor. Therefore, the long-term effect of environmental regulation on economic development cannot depend only on technological innovation by coal enterprises themselves. Rather, it should be implemented to improve the production efficiency of coal enterprises and accelerate intelligent construction. The long-term effect of economic development on environmental regulation needs to bring environmental regulation output through structural upgrades and technological progress [17]. According to the "Compliance Costs Theory", environmental pollution has externality [14]. To improve social welfare, the state's environmental regulation investment accounts for a certain amount of industry investment, and coal enterprises also need to pay a certain amount of the pollution control costs. However, the production costs of coal enterprises increase with the increase of pollution control costs. If the technical situation and demand conditions remain unchanged, it will inevitably lead to a decline in productivity and profit margin, which will inhibit economic development [18].

Coal mine accidents inhibit economic development because accidents lead to death, equipment damage, mine shutdown, resource damage and other consequences, causing serious economic losses [19]. Economic development has a long-term inhibitory effect on coal mine accidents. According to the theory of safety economics, when the economic growth rate drops and the social system is stable, the accident rate drops significantly and tends to be stable. At present, China is at this stage, so economic growth has played a restraining role in coal mine accidents. According to the theory of safety economics, stable economic development will inevitably reduce the probability of coal mine accidents, but as

long as accidents occur, economic losses will inevitably result [3]. Therefore, we can only control the occurrence of coal mine accidents, not eliminate them. In this regard, the impact of coal mine accidents on economic development in China's ten major coal provinces is relatively good.

## 6. Conclusions

On the basis of combing the existing research literature, this paper builds a PVAR model based on the provincial panel data of ten major coal provinces in China from 2011 to 2020 and conducts empirical research on the interaction between coal mine accidents, environmental regulation and economic development. The specific conclusions are as follows:

(1) From the static development between coal mine accidents, environmental regulation and economic development in China's ten major coal provinces, the three variables have achieved coordinated development. Only the impact mechanism of environmental regulation on economic development is unreasonable. The impact coefficient of coal mine accidents on environmental regulation is 0.04; coal mine accidents promote environmental regulation. The impact coefficient of environmental regulation on coal mine accidents is $-0.176$. Environmental regulation inhibits coal mine accidents. The impact coefficient of economic development on environmental regulation is 0.791; economic development strongly promotes environmental regulation. The impact coefficient of environmental regulation on economic development is $-0.0015$; environmental regulation has a weak inhibitory effect on economic development. The impact coefficient of coal mine accidents on economic development is $-0.015$; coal mine accidents slightly inhibit economic development. The impact coefficient of economic development on coal mine accidents is $-1.035$; economic development strongly inhibits coal mine accidents.

(2) The dynamic development relationship between the three variables, coal mine accidents, environmental regulation and economic development, is good as a whole but needs to be optimized. There is a strong dynamic response relationship between environmental regulation and coal mine accidents. When environmental regulation inhibits (promotes) coal mine accidents, the accidents will promote (inhibit) environmental regulation, but on the whole, the impact of environmental regulation on accidents shows a decreasing volatility trend, and the impact of coal mine accidents on environmental regulation shows a rising volatility trend. There is a short-term positive interaction between economic development and environmental regulation, but the interaction relationship between them decreases with time. The dynamic development relationship between coal mine accidents and economic development has been relatively reasonable. Ten major coal provinces have shown that economic development has a long-term inhibitory effect on coal mine accidents, which conforms to the theory that stable economic development will inhibit coal mine accidents. The negative impact of coal mine accidents on economic development has gradually decreased to 0.

**Author Contributions:** Conceptualization, C.L.; writing—original draft preparation, C.L. and S.L.; writing—review and editing, C.L., S.L., K.X. and J.L.; supervision, S.L. All authors have read and agreed to the published version of the manuscript.

**Funding:** This study was supported by the National Natural Science Foundation of China (71972176) and Major projects of Jiangsu Social Science Foundation of China (21ZD006).

**Institutional Review Board Statement:** Not applicable.

**Informed Consent Statement:** Not applicable.

**Data Availability Statement:** The Microsoft Excel Worksheet data used to support the findings of this study are available from the corresponding author (lishuangchina@cumt.edu.cn) upon request.

**Conflicts of Interest:** The authors declare no conflict of interest.

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
