# Peer review of "Coal Mine Safety Accidents, Environmental Regulation and Economic Development—An Empirical Study of PVAR Based on Ten Major Coal Provinces in China"

_sustainability, doi:10.3390/su142114334_

Round 1

Reviewer 1 Report

This manuscript studied the dynamic relationship between coal safety accidents, environmental regulation and economic development with the PVAR model, which is interesting and novel. The figures in the manuscript should be redrawed with scientific drawing software, such as matlab, origin.

Author Response

Point 1: This manuscript studied the dynamic relationship between coal safety accidents, environmental regulation and economic development with the PVAR model, which is interesting and novel. The figures in the manuscript should be redrawed with scientific drawing software, such as matlab, origin.

Response 1: Thanks for the reviewer’s kind advice. We've redrawn all the figures. Details are as follows.

Reviewer 2 Report

Although the paper analyzes a topic of interest, there are several weaknesses that have to be addressed before its publication.  They are discussed below.

The proposed model seems too linear and does not adequately take into account the complexity of the operational environment (there are new hazards and risks related to the complexity of operational environment and organizational and human performance) as well as the influence of biases in decision making (motivational and cognitive) leading to accidents. Safety culture/management role such as presented in the model do not capture it well in this new context.

The scope of analysis is not sufficient. Such an approach often leads to putting the blame on frontline workers for deficiencies which are at the organizational level enabling and tolerating conditions creating unsafe workplaces (leading to the “drift to failure/accident”). The practice shows that these influences are “soft factors” that are hard to resolve. Recent research works and return of experience show that the organizational performance plays a key role in creating conditions for accidents. The organizational performance also includes less studied motivational biases in decision-making process (mentioned above), and it is not considered either (cognitive biases are relatively well studied and understood).

Thus, the scope of the paper does not enable to capture the true image of the analyzed topic. Given the limited scope of the paper, the literature review is also too narrow, and it should be expanded to include aspects discussed above.

It is strongly recommended that the authors consult the following references while revising the paper.

Dekker S, Cilliers P, Hofmeyr, J.H., (2011), The complexity of failure: Implications of complexity theory for safety investigations. Safety Science. 49: 939-945

Komljenovic, D., Loiselle, G., Kumral, M., (2017), Organization: a new focus on mine safety improvement in a complex operational and business environment, International Journal of Mining Science and Technology, 27: 617-625

Leveson N.G., (2011b), Applying system thinking to analyze and learn from events. Safety Science, 49:55-64

Montibeller, G. and Winterfeldt, D. Cognitive and Motivational Biases in Decision and Risk Analysis, Risk Anal. 2015: 35 (7): 1230-1251

Author Response

Point 2: The proposed model seems too linear and does not adequately take into account the complexity of the operational environment (there are new hazards and risks related to the complexity of operational environment and organizational and human performance) as well as the influence of biases in decision making (motivational and cognitive) leading to accidents. Safety culture/management role such as presented in the model do not capture it well in this new context.

The scope of analysis is not sufficient. Such an approach often leads to putting the blame on frontline workers for deficiencies which are at the organizational level enabling and tolerating conditions creating unsafe workplaces (leading to the “drift to failure/accident”). The practice shows that these influences are “soft factors” that are hard to resolve. Recent research works and return of experience show that the organizational performance plays a key role in creating conditions for accidents. The organizational performance also includes less studied motivational biases in decision-making process (mentioned above), and it is not considered either (cognitive biases are relatively well studied and understood).

Thus, the scope of the paper does not enable to capture the true image of the analyzed topic. Given the limited scope of the paper, the literature review is also too narrow, and it should be expanded to include aspects discussed above.

It is strongly recommended that the authors consult the following references while revising the paper.

Response 2: Thanks for the reviewer’s question. First of all, Komljenovic believes that organizational performance is an important micro factor affecting coal mine safety accident [1], and Wang believes that operational environment, safety culture/management role are important micro factors affecting coal mine safety accident [2], which is consistent with your opinion. In this way, if we put those variables into the model, it will cause endogenous and collinearity problems of variables, which will adversely affect the final results of the model. Most importantly, our model contains random error term , which involve the influence of microscopic variables.

Therefore, we think your opinion is very important, but considering the operability of the model, we cannot make in-depth modifications to it. Thank you again for your professional guidance.

[1] Komljenovic, D., Loiselle, G., Kumral, M., (2017), Organization: a new focus on mine safety improvement in a complex operational and business environment, International Journal of Mining Science and Technology, 27: 617-625

[2] Wang, L., Cao, Q., & Zhou, L. (2018). Research on the influencing factors in coal mine production safety based on the combination of DEMATEL and ISM. Safety science, 103, 51-61.

Reviewer 3 Report

This paper analyzes the relationship among Coal Mine Safety Accidents, Environmental Regulations and Economic Development. I suggest that this paper is acceptable after revision. Please read the following questions carefully and make answers and modifications.

1.                  Please adjust the paragraph structure of the introduction part to make it logical and clear.

2.                  Please write down the references of the last sentence of the first paragraph of Part 2.1, the second sentence of the first paragraph of Part 2.2,

3.                  I'd like to see more analysis of the data itself, and conclusions based on that. I mean, your current analysis is not specific and highly subjective.

4.                  I would like you to place the individual line charts in Figure 2 near the corresponding paragraphs so that they are easy to read.

5.                  Please adjust the picture to the right size and pay attention to the sharpness of the picture.

6.                  Please increase the proportion of data analysis in your paper to make it more convincing.

7.                  Please readjust the sections and I hope your innovations and discoveries take more space.

8.                   Please review the format of the article, for example, the format of chart titles

9.                   Please readjust your writing to make it more in line with the English reading habits.

Author Response

Point 1: Please adjust the paragraph structure of the introduction part to make it logical and clear.

Response 1: Thanks for the reviewer’s questions. We have re adjusted the introduction to four paragraphs.

Point 2: Thanks for the reviewer’s questions. Please write down the references of the last sentence of the first paragraph of Part 2.1, the second sentence of the first paragraph of Part 2.2.

Response 2: We have added references to the last sentence of the first paragraph of Part 2.1 and the second sentence of the first paragraph of Part 2.2.

Point 3: I'd like to see more analysis of the data itself, and conclusions based on that. I mean, your current analysis is not specific and highly subjective.

Response 3: Thanks for the reviewer’s questions. We have made more analysis on the data, and based on this, we have modified the conclusion and abstract. Details are as follows.

Point 4: I would like you to place the individual line charts in Figure 2 near the corresponding paragraphs so that they are easy to read.

Response 4: We have split the broken line charts in Figure 2 into four line charts and placed them near the corresponding paragraphs.

Point 5: Please adjust the picture to the right size and pay attention to the sharpness of the picture.

Response 5: Thanks for the reviewer’s suggestion. We have readjusted the size and sharpness of all the figures in the manuscript.

Point 6: Please increase the proportion of data analysis in your paper to make it more convincing.

Response 6: Thanks for the reviewer’s suggestion. We have split the main data analysis chapter 4.3 in the manuscript into 4.3 and 4.4, deleted the discussion content and supplemented the data analysis content. Details are as follows.

Point 7: Please readjust the sections and I hope your innovations and discoveries take more space.

Response 7: Thanks for the reviewer’s suggestion. We have readjusted the distribution of chapters in the manuscript, modified the content as a whole, and supplemented the innovation and discovery of the article

Point 8: Please review the format of the article, for example, the format of chart titles.

Response 8: Thanks for the reviewer’s suggestion. We reviewed the format of the manuscript and corrected all formatting errors.

Point 9: Please readjust your writing to make it more in line with the English reading habits.

Response 9: Thanks for the reviewer’s suggestion. We rechecked all English writing in the manuscript and made minor modifications.

Round 2

Reviewer 2 Report

In their response 2, the authors state: First of all, Komljenovic believes that organizational performance is an important micro factor affecting coal mine safety accident [1], and Wang believes that operational environment, safety culture/management role are important micro factors affecting coal mine safety accident [2], which is consistent with your opinion.

Comment by the reviewer: the organizational performance, environment and safety culture are not micro factors. Historical data in many industries at risk (including mining) show that those are major even dominant factors, and cannot be excluded from any model aiming at analyzing safety issues.

In this way, if we put those variables into the model, it will cause endogenous and collinearity problems of variables, which will adversely affect the final results of the model. Most importantly, our model contains random error term, which involve the influence of microscopic variables.

Therefore, we think your opinion is very important, but considering the operability of the model, we cannot make in-depth modifications to it. Thank you again for your professional guidance.

Comment by the reviewer: I disagree. As per response, it seems that the authors aim at fitting the reality into the model, and not developing the model which reflects the reality. It is fundamentally wrong and has to be corrected.

[1] Komljenovic, D., Loiselle, G., Kumral, M., (2017), Organization: a new focus on mine safety improvement in a complex operational and business environment, International Journal of Mining Science and Technology, 27: 617-625

[2] Wang, L., Cao, Q., & Zhou, L. (2018). Research on the influencing factors in coal mine production safety based on the combination of DEMATEL and ISM. Safety science, 103, 51-61.

Author Response

Thanks for the reviewer’s question. We fully agree with you that organizational performance, safety culture and organizational environment may affect coal mine safety accident as intermediary variables, but the empirical research should be extensive. In the literature review, we fully discussed the possible relationship between coal mine safety accident, environmental regulation and economic development, and through empirical research, we proved that the research results are in line with the actual development trend, Our research is certainly of practical and theoretical significance.

In addition, Shi explored the impact of government regulations on coal mine safety accident[1], and Bai discussed the impact of the policy of closing the mine on coal mine safety accident[2]. These scholars did not consider the impact of organizational performance, environment and safety culture on coal mine safety accident. We believe that the manuscript mainly studies the dynamic development relationship between coal mine safety accident, environmental regulation and economic development, and grasps the macro development trend of coal mine safety accident, environmental regulation and economic development, rather than exploring the main factors affecting coal mine safety accident. If we consider the impact of organizational performance, safety culture and organizational environment on coal mine safety accident. Then the manuscript should be changed to the research on the influencing factors of coal mine safety accident, or to the research on the influence of organizational performance, safety culture and organizational environment on coal mine safety accident.

[1] Shi, X. (2009). Have government regulations improved workplace safety?: a test of the asynchronous regulatory effects in China's coal industry, 1995–2006. Journal of safety research, 40(3), 207-213.

[2] Bai, C.E.; Wang, X.; Zhong, X.H. Regulation and Property Right:Effect of China's Coalmine Shutdown Policy on Work Safety. China Soft Science, 2011, 12-26.

Reviewer 3 Report

After modification, it meets the recruitment standard, and it is recommended to accept

Author Response

Thank you again for your recognition and your work on our manuscript.

Round 3

Reviewer 2 Report

Agree with the newest revision of the submission.